# EvalRes: Evaluating VLMs' Sensitivity to Image Resolution and Relative Detail Size

## Abstract

Visual Language Models (VLMs) have achieved remarkable success across a wide range of Visual Question Answering (VQA) tasks. Yet, they still struggle with high-resolution visual inputs where regions providing key information are relatively small or scenes are highly detailed and cluttered. This limitation stems from the architectural bottlenecks of current vision encoders, which often fail to preserve fine-granular details necessary for precise reasoning. While several approaches have been proposed to address this issue, a systematic evaluation of a model's capacity to process high-resolution content and small-scale visual cues has been lacking. In this work, we introduce a versatile framework to extend benchmarks and propose two novel metrics designed to assess VLMs' scalability across varying image resolutions and aspect ratios. Unlike evaluation with existing benchmarks, which lack consistency in image properties and fail to isolate resolution and aspect ratio effects, our method enables controlled experimentation to disentangle resolution sensitivity from the overall task performance. Our framework not only enables more robust and fair VLM evaluation, but also paves the way for future research into high-fidelity visual understanding. We evaluate several widely used VLMs with the proposed framework, revealing that even state-of-the-art models struggle with higher resolution and non-standard aspect ratios, and that processing small details remains a major challenge.

## 1 Introduction

The ability to process fine-grained visual information is essential for a wide range of VLM applications, that deal with medical and satellite imaging, large diagrams or plans, or involve recognizing texts, charts, and performing mathematical reasoning. For all these tasks, critical visual cues often reside in small, high-resolution regions of an image. At the same time, most existing architectures, such as Vision Transformers (ViTs), are constrained by computational and memory limitations that force them to process images at relatively low resolutions (e.g., 224×224 or 384×384 pixels), leading to significant information loss when dealing with high-resolution inputs.

Currently existing benchmarks evaluate a variety of skills and abilities and often include images that differ in size and aspect ratio. As we will demonstrate in this paper, even current state-of-the-art models still break down at higher resolutions. However, the degree to which this occurs depends primarily on the design of the vision encoder. This may undermine evaluations of other model abilities, such as reasoning and chain-of-thought, as in some cases wrong answers are caused by information loss due to bottlenecks in image encoding rather than a lack of reasoning. In this paper, we aim to disentangle performance effects that are due to an inability to scale and process images of higher resolution or non-standard aspect ratios from other abilities of interest.

In summary, our contributions are threefold:

- We propose a novel way to extend any VQA benchmark to account for a model's ability to work with small details, higher resolutions, and different aspect ratios.

- We assess our benchmark extension method across several commonly used VQA models and benchmarks, systematically categorize the failure modes and conduct thorough experiments to measure the effects on various VQA domains.

- We propose two novel metrics that enables quantitative comparisons of models based on their ability to scale to higher resolutions and unconventional aspect ratios.

## 2 RELATED WORK

The issue of resolution-induced information loss has been highlighted in various domains. For example, high-resolution input is essential in autonomous driving, where small objects and distant signs must be detected (Zhou et al., 2025). The same applies to OCR and document understanding (Nacson et al., 2024) as well as for general-domain VQA, where the visual cues for answering the question may be small (Shi et al., 2025).

In this section, we review recent approaches aimed at enhancing the ability of VLMs to process high-resolution visual inputs. To align with our objective, we first examine methods that improve VLM scalability to high-resolution images and support for flexible aspect ratios—particularly in scenarios where critical visual information is sparse or localized. We then analyze existing visual question answering benchmarks, emphasizing how they evaluate core VLM capabilities and identifying their limitations in isolating resolution-related effects. Finally, we survey approaches to measuring performance degradation, as these are essential for understanding the impact of resolution on model behavior.

### 2.1 HIGH-RESOLUTION IMAGE PROCESSING AND VISUAL HAYSTACK PROBLEMS

Most current state-of-the-art VLMs use vision encoders based on the ViT architecture, typically trained using CLIP-style pretraining. However, both the original ViT and subsequent models such as CLIP, SigLIP, and BLIP rely on a fixed image resolution during training (Dosovitskiy et al., 2021). In CLIP and SigLIP, for instance, images are resized to a constant size and then center-cropped (Radford et al., 2021; Zhai et al., 2023). Similarly, BLIP applies a random crop to a subregion of the input image, followed by bicubic resizing. While effective for general-purpose training, these fixed-resolution pipelines inherently discard fine-grained visual details (Li et al., 2022).

While traditionally extending the ability of a ViT to work with higher-resolution images was achieved by fine-tuning the vision encoder with higher-resolution inputs (Touvron et al., 2019), this method is limited by high computational costs and is not robust to varying input sizes.

More and more VLMs aim to process images at native resolution without resizing or cropping and address the resolution challenge more scalably. Thus, Li et al. (2024); Liu et al. (2024a); Xu et al. (2024) propose partitioning high-resolution images into smaller patches (tiles) that match ViT's native input size and processing them independently together with a wholly-encoded global image. This allows models to maintain a consistent input format while still accessing high-detail regions. However, this method still puts constraints on the maximum resolution and the computational costs are high. A more recent alternatives involve selectively identifying and processing only the task-relevant portions of the image at high resolution, significantly reducing computational overhead while preserving critical visual content (Zhou et al., 2025; Shi et al., 2025). Another research direction explores idea of adaptive image tokenization based on visual complexity. For example, Shen et al. (2025) proposes to dynamically adjust the token density in accordance with local image content in contrast to using fixed-size image patches, thus preserving detail in visually rich areas while saving computation elsewhere.

### BENCHMARKS

Since the emergence of the first large VLMs capable of advanced reasoning and visual understanding, a wide variety of benchmarks have been developed to evaluate different dimensions of VLMs' capabilities. Significant effort has gone into designing comprehensive, multidimensional evaluations of VLMs (Gupta et al., 2022; Yu et al., 2024a;b; Liu et al., 2024b).

In addition to these broad evaluations, other benchmarks focus on assessing specific abilities, such as complex reasoning skills (Han et al., 2023b;a), susceptibility to visual hallucinations (Guan et al., 2023; Wu et al., 2024; Guan et al., 2024), or detecting the boundaries of commonsense reasoning (Bitton-Guetta et al., 2023). Significant attention has been devoted to evaluating VLMs' performance with schematic information, such as diagrams (Lu et al., 2021), flowcharts (Singh et al.,

2024), and tables (Kim et al., 2024). Sun et al. (2024) study abilities that involve more complex chains of thought by evaluating the process and outcome of VLMs solving middle school math problems with visual contexts.

More recently, increasing attention has been directed toward evaluating VLMs' ability to process long visual contexts and address the "Visual Haystack" problem. For example, Wu et al. (2025) examine whether models can locate critical visual information within a large pool of images and subsequently use it to answer specific queries. Shi et al. (2025) proposed a new high-resolution benchmark that contains QA pairs with questions that can be answered only at 4K resolution. However, these benchmarks do not systematically study the scalability at different resolutions. Niu et al. (2025) propose a new resolution-centric benchmark specifically tailored for evaluating visual encoding strategies under extreme visual conditions (encompassing various resolutions and aspect ratios).

## 2.2 MODEL ROBUSTNESS METRICS.

The degradation of VLMs' performance stemming from the loss of information due to image downsizing and limited numbers of visual tokens were highlighted by several studies (Zhou et al., 2025; Dehghani et al., 2023).

Currently, in order to evaluate effectiveness of new approaches aimed to improve model's scalability and robustness to higher resolutions and different aspect ratios, it is a common practice to test them on datasets containing images of diverse resolutions (Dehghani et al., 2023; Tan et al., 2025; Xu et al., 2024). While this approach allows observing general model performance improvement on a diverse set of VQA questions from diverse domains, it, however, does not isolate or quantify effects due to the model's scalability and robustness to resolution from other skills and abilities. Moreover, these benchmarks contain information with heterogeneous levels of detail and granularity, which hinders a fair comparison of the models.

To our knowledge, there is currently no prior work that has systematically analyzed the impact of resolution increase on VLMs' performance and limits of their scalability in isolation from other VLM performance aspects.

## 3 CAN VLMS HANDLE HIGH-RESOLUTION VISUAL INPUTS WITH SMALL DETAILS?

To answer this question, we propose an experimental setup that allows to test how model performance reacts to changes in image resolution and the size of important details. Using proposed method, we evaluate several recent commonly used VLMs on a range of specialized VQA benchmarks, designed to evaluate different capabilities. Our results demonstrate that even the most advanced VLMs suffer from notable performance degradation at higher resolutions, as their VQA performance drops drastically with an increase in image size. In the following, we provide details on our experimental settings to assess VLMs' ability to scale to higher resolutions and work with small details, along with the evaluation protocol and finally the experimental results.

### 3.1 METHOD

Conceptually, we aim to design a setup in which a set of images contains the same information content but varies in resolution and relative detail size. There are multiple ways to achieve this.

One approach is image upscaling (e.g. using bi-linear or bi-cubic interpolation). However, such methods alter the image and the relationships between pixels, and all important details are scaled proportionally with the image size. Another alternative is to use generative methods, such as upscaling or outpainting, to increase image resolution. While these methods can produce high-resolution images, they are computationally expensive. Furthermore, they remain difficult to control and may inadvertently introduce new details that could change the right answer.

In this paper, we propose a simple method that satisfies the desired properties and can be applied to any benchmark dataset. The key idea is to augment existing images with neutral backgrounds of varying sizes, ensuring that the original answer remains unaffected and no additional information is added (see Figure 1).

Our experiments, described in the following sections, show that even the best-performing models currently available struggle with this seemingly straightforward test.

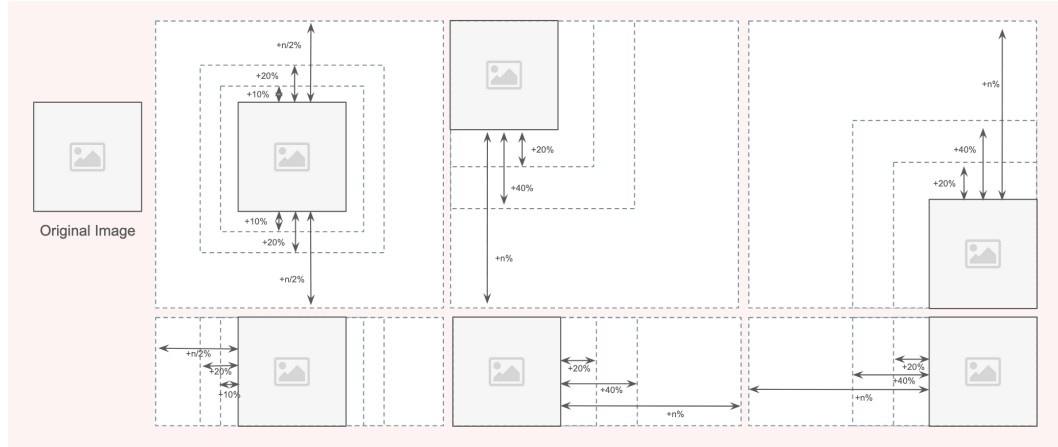

Figure 1: Method Illustration: The original image is placed on a neutral background (e.g., white with RGB values (255, 255, 255)) of varying sizes. Depending on the experimental goals, the background can extend the image in a single direction—altering the aspect ratio—or in both directions. Additionally, we recommend testing multiple image positions (e.g., top-left corner, center, bottom-left corner) to evaluate the model's sensitivity to the spatial location of key visual cues.

## 3.2 SCALABILITY AND ROBUSTNESS EVALUATION METRICS

To our knowledge, no dedicated quantitative metric exists to assess a model's scalability with respect to resolution alone, which obscures the causes of accuracy drops and makes comparisons across diverse models unfair. We therefore introduce two complementary metrics to quantify vision encoders ability to scale.

### 3.2.1 AREA UNDER THE SCALING CURVE (AUSC)

First, we introduce the Area Under the Scaling Curve (AUSC), a metric that quantifies how well models adapt to varying resolutions and aspect ratios, enabling fair comparison of VLMs independent of architecture, size, or overall performance.

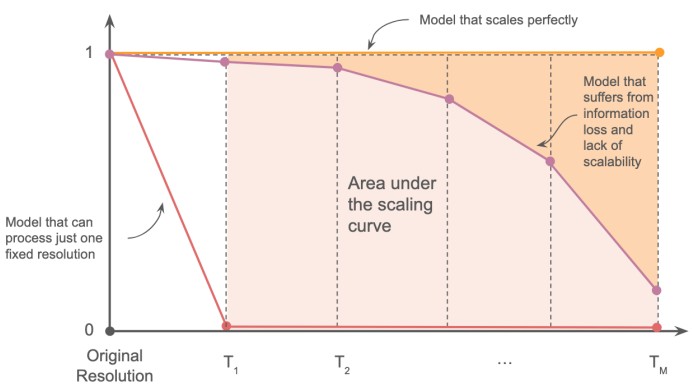

Figure 2: Illustration of the Area Under the Scaling Curve (AUSC) metric

To make AUSC values comparable for VLMs regardless of their overall performance, it is calculated on the subset of the original benchmark for which the model provides correct answers for the original image resolution. Let $\mathcal{B}$ be a chosen benchmark that consists of $(x_i, y_i)$ image–question pairs and

ground truth answers $(z_i)$, and let $f(x)$ denote a chosen VLM. Then the subset of image-question pairs from the benchmark $\mathcal{B}_{correct} \subseteq \mathcal{B}$ for that the model provides the correct answer is defined as:

$$\mathcal{B}_{correct} = \{(x_i, y_i) \in \mathcal{B} \mid f(x_i, y_i) = z_i\} \tag{1}$$

Let $\mathcal{T} = \{T_1, T_2, \ldots, T_M\}$ be a set of image transformations as described in the previous section, where $M$ is a size of the added white margin with respect to the original image size. Then a transformed benchmark subset can be defined as:

$$\mathcal{B}_{correct}^{(j)} = \{(T_j(x_i), y_i) \mid (x_i, y_i) \in \mathcal{B}_{correct}\} \tag{2}$$

For each size of the margin $j = 1, 2, ..., M$, we evaluate the model $f(x)$ based on the chosen metric and evaluation protocol (for example, VQA accuracy). We denote this set of experiments $\mathcal{E}$ for each of the image transformations in $\mathcal{T}$ as:

$$\mathcal{E} = \{(T_j, \mathrm{Acc}(T_j)) \mid j = 1, \ldots, M\} \tag{3}$$

where $\mathrm{Acc}(T_j)$ is the VQA accuracy on $\mathcal{B}_{correct}$ that is defined as

$$\mathrm{Acc}(T_j) = \frac{1}{|\mathcal{B}_{correct}|} \sum_{(x_i, y_i) \in \mathcal{B}_{correct}} \mathbb{1}_{correct}\big[f(T_j(x_i), y_i) = z_i\big] \tag{4}$$

,

where $\mathbb{1}_{correct}$ is an indicator function. Given $\mathcal{E}$, we can construct a model scaling curve and calculate the area under that curve as:

$$\mathrm{AUSC} = \frac{1}{M-1} \sum_{j=1}^{M-1} \frac{1}{2}\bigg[\mathrm{Acc}(T_j) + \mathrm{Acc}(T_{j+1})\bigg] \tag{5}$$

AUSC values lie in $[0, 1]$ and have an intuitive interpretation:

- **AUSC $= 1$**: The model scales perfectly under the set of transformations $\mathcal{T}$. This indicates *maximum robustness*.
- **AUSC $\in (0, 1)$**: Model architecture and image preprocessing steps incur visual information loss. Higher AUSC values reflect smoother degradation and more stable performance across transformations. Lower AUSC values indicate that the model is sensitive to certain transformations and quickly degrades.
- **AUSC $= 0$**: The model can process just one fixed resolution and fails to recognize any information in images of higher resolutions. This indicates *complete brittleness*.

One of the limitations of the AUSC is the size of $\mathcal{B}_{correct}$. For the purpose of statistical significance, we recommend $|\mathcal{B}_{correct}| \geq 75$.

### 3.2.2 RESOLUTION BIAS SCORE (RBS)

For additional insight, we further assess how variable the predictions are under the set of transformations $\mathcal{T}$ when considering instances that the model originally did not answer correctly. In order to measure this, we propose calculating a Resolution Bias Score (RBS):

$$\mathrm{RBS}(T_j) = \frac{1}{|\mathcal{B}_{wrong}|} \sum_{(x_i, y_i) \in \mathcal{B}_{wrong}} \mathbb{1}_{correct}\big[f(T_j(x_i), y_i) = z_i\big] \tag{6}$$

This metric captures how often the transformation $T_j$ causes the model to produce the correct answer on an image–question pair for which the model produced an incorrect prediction in the original resolution. We later show that this happens at a surprisingly high rate for certain models.

Table 1: AUSC for different positions of visual cues

|  | **Overall** | **Center** | **Top** | **Bottom** |
|---|---|---|---|---|
| Llama-3.2 | 0.654 | 0.674 | 0.612 | 0.675 |
| Qwen-2.5 | 0.645 | 0.661 | 0.643 | 0.630 |
| LlaVa-1.5 | 0.551 | 0.646 | 0.392 | 0.401 |
| Pixtral | 0.632 | 0.659 | 0.610 | 0.631 |
| GPT-4o | 0.703 | 0.712 | 0.688 | 0.709 |

The RBS lies in $[0, 1]$, where 0 means that the model is invariant towards shifts and size changes of visual information, while the maximum score of 1 implies extreme sensitivity towards such changes. Based on the chosen experiment transformations set $\mathcal{T}$, the outcomes of the analysis and interpretation of the metric may differ. High values of the RBS may signal:

- Potential data augmentation benefits for vision encoder training
- Shifts in model attention or reasoning, which in turn means a lack of vision encoder robustness towards image transformations

Similarly to AUSC, RBS values are dependent on the size of the subset it is calculated on. Therefore, to ensure significance and comparability of the results, we recommend to choose sufficiently large and challenging benchmarks with $|\mathcal{B}_{\text{wrong}}| \geq 75$.

## 4 EXPERIMENTAL RESULTS AND ANALYSIS

In this section, we discuss the setup choices and the results obtained by applying our method to compare several commonly used VLMs.

### 4.1 EXPERIMENTAL SETUP

To assess differences in scaling across different types of VQA, we chose MMVet (Yu et al., 2024a), CharXiv (Wang et al., 2024), MME (Fu et al., 2024), MMStar (Chen et al., 2024), and MathVerse (Zhang et al., 2024) benchmarks for further analysis.

For all the images in the benchmarks, we assess:

- 9 added margin sizes (20%, 40%, 60%, 80%, 100%, 200%, 300%, 400%, 500%). The choice of the margins is motivated by our desire to separately analyse the effects of small ($\leq$100%) and large margins ($\geq$100%) on overall model performance as well as how each of the different VQA types is affected.
- 3 positions of the image within the added neutral background, in order to understand how the position of critical visual cues affects model performance.
- 2 margin configurations according to the method described in Section 3.1 to disentangle sensitivity to resolution and to aspect ratio.

For our analysis, we chose several commonly used open and closed source models, specifically LLaVa-1.5 (Liu et al., 2024a), Llama-3.2 (Grattafiori et al., 2024), Qwen-2.5-3B (Xu et al., 2025), Pixtral (Agrawal et al., 2024) and GPT-4o (OpenAI, 2024).

### 4.2 RESOLUTION SENSITIVITY ANALYSIS

Our results demonstrate that it is still a challenge for the commonly used VLMs to process large, high-resolution images with small details. For all analyzed VLMs, we observed significant performance degradation with image size increase, in particular, for margin sizes that more than double (extra100%) each of the original image dimensions (See Figure 3).

Interestingly, LLaVa1.5 demonstrated superior scaling capabilities for the image placed in the center, while its performance dropped close to 0 for all other visual cues positions. This illustrates

cropping based nature of the vision encoder implemented in LLaVa1.5, which in other tested models is replaced with more advanced designs. For the rest of the tested models, we did not observe any significant differences in the performance due to the position of critical visual cues (Figure 4). We observed similar effects for changing the aspect ratio.

To quantitatively compare performance degradation across analysed VLMs, we calculated proposed AUSC and RBS metrics. The results in Table 1 and Table 3 illustrate that regardless of the original performance on the benchmark, VLMs share a broadly similar ability to scale, although bigger models tend to scale slightly better.

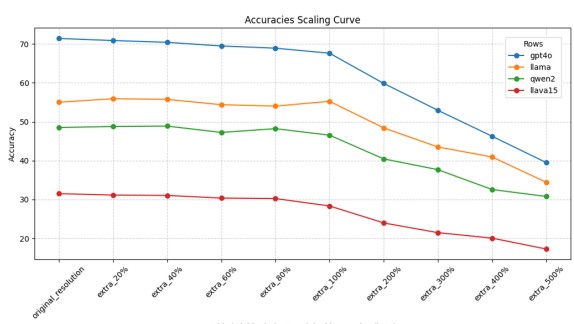

Figure 3: Accuracy degradation as the image size increases for MMVet Benchmark (averaged across different positions)

Based on the RBS values, we conclude that predictions of all tested VLMs are highly sensitive to the shifts and scale changes of the critical visual cues. We observe that LLaMA-3.2 and Qwen-2.5 exhibit notably high variance in their predictions under the proposed set of transformations (see Table 3). Interestingly, even modest padding—margins that increase the image dimensions by less than 100% and have minimal impact on average performance—can lead to significant shifts in the predicted outputs. For both LLaMA-3.2 and Qwen-2.5, over half of the answers that were initially incorrect on the original image became correct after the input transformation. These findings indicate that the models lack invariance to spatial transformations: slight alterations in the positioning or scale of visual content, despite adding no new information, can substantially affect model predictions. Interestingly, LLaVa model, being relatively concise when providing correct answers, goes into reasoning loops faced with higher uncertainty and inability to detect information relevant to provide correct answer. On average, we noticed that wrong answers teng to be longer than correct ones, signaling models overcompensating for uncertainty with additional elaboration.

To gain further insights into the reasons of discovered performance degradations and answer instabilities, we attempted to identify patterns in how exactly answers change due to the input transformation. One interesting pattern we noticed, is the compensatory elaboration in the answers (e.g. Figure 5). We observed that models differ a lot in their reaction to higher input size. While Qwen's responses become more concise, LLaMa compensates for increased uncertainty with additional elaboration which results into significantly longer responses.

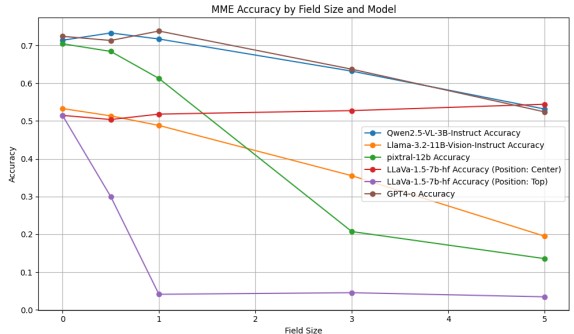

Figure 4: Accuracy degradation as the image size increases for MME Benchmark

Closely looking at the results on different specialized benchmarks, we could also notice that model sensitivity to resolution is dependent on the type of the VQA task. While some instances require a general understanding of the overall image based on large visual cues, tasks such as OCR, mathematical reasoning, table understanding from images, and diagram understanding require being able to extract a large number of small details (see Appendix B Figure 6).

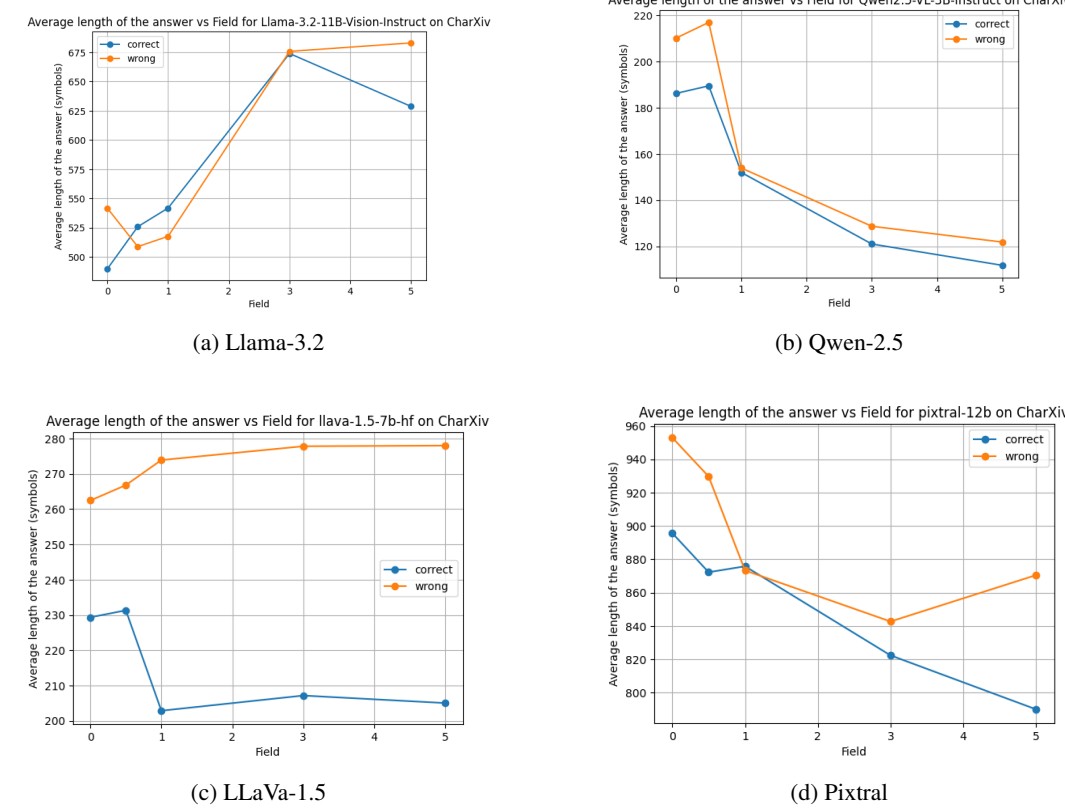

Figure 5: Dependence of VLMs' responce length on the size of visual input (CharXiv benchmark)

Table 2: RBS across varying margin sizes and visual cue positions measured on MMVet Benchmark

| Model | Margin Sizes | Overall | Center | Top | Bottom |
|---|---|---|---|---|---|
| Llama-3.2 | all | 0.607 | 0.622 | 0.600 | 0.600 |
| Qwen-2.5 | | 0.581 | 0.581 | 0.581 | 0.588 |
| LlaVa-1.5 | | 0.222 | 0.222 | 0.222 | 0.289 |
| GPT-4o | | 0.423 | 0.423 | 0.462 | 0.404 |
| Llama-3.2 | large ($> 100\%$) | 0.378 | 0.378 | 0.444 | 0.489 |
| Qwen-2.5 | | 0.459 | 0.459 | 0.453 | 0.514 |
| LlaVa-1.5 | | 0.222 | 0.222 | 0.200 | 0.244 |
| GPT-4o | | 0.295 | 0.308 | 0.269 | 0.308 |
| Llama-3.2 | small ($< 100\%$) | 0.559 | 0.567 | 0.578 | 0.533 |
| Qwen-2.5 | | 0.543 | 0.554 | 0.541 | 0.534 |
| LlaVa-1.5 | | 0.219 | 0.189 | 0.211 | 0.256 |
| GPT-4o | | 0.391 | 0.365 | 0.423 | 0.385 |

## 5 CONCLUSION

Our paper presents the first benchmark to systematically study resolution scalability in isolation, disentangled from other challenges of VLMs. To this end, we introduce a flexible framework to induce benchmark datasets along with novel evaluation metrics to assess the scaling behavior.

Our experimental evaluation yielded three key insights into VLM behaviors on high-resolution images with small details and visual reasoning tasks: (1) modern VLMs suffer from rapid performance deterioration at more than double resolution extension, moreover VLMs tend to compensate for

higher uncertainty to longer and more elaborate answers (not necessarily correct at the end), (2) vision encoders are very sensitive to small image shifts and size changes. In contrast, the position of critical visual cues in the image usually has negligible effects (3) VLMs' sensitivity to changes in the input resolution are highly dependent on the VQA type. Unsurprisingly, VLMs exhibit greater sensitivity on tasks that require detailed understanding of small details hidden in the image, e.g., OCR and mathematical problems. We further found that regardless of original performance on the benchmark, closed-source GPT-4o outperforms the considered open-source VLMs in both the scalability and stability of the visual encoder.

Overall, this study hence lays the groundwork to facilitate new research on learning better visual representations that are invariant towards input transformations and induce less information loss when processing higher resolutions visual inputs.

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

## A  LIMITATIONS AND FUTURE WORK

In the previous sections, we demonstrated that our method provides valuable insights into model robustness and scalability and is designed to be broadly applicable for uncovering model limitations across diverse application scenarios. In this section, we explicitly highlight the limitations of our current analysis and outline promising directions for future research.

As with any multifold evaluation approach, our method requires running model inference multiple times — depending on the selected set of evaluation margins. As such, for large-scale in-depth analysis, we suggest using small- to medium-sized benchmarks or computing metrics on a representative subsample to ensure computational feasibility while maintaining statistical validity. In addition, in this study we disabled sampling feature of the tested VLMs to make responses more stable across runs. However there are still sources of stochasticity that may substantially affect stability of model responses. We plan to incorporate this analysis in our future work.

While not always possible working with existing models, to foster deeper understanding of performance degradation and model sensitivity to shifts and rescaling of important details, we recommend analyzing attention maps of the model for the cases when answer correctness switches.

Our analysis uncovered that even the best currently existing models suffer from visual information loss due to striking limitations of the visual context length they can process, architectural bottlenecks such as lack of invariance towards shifts and scaling of visual cues, and attention fragmentation. As this is critical for many applications, we highlight the necessity of further research in this area. Using proposed experimental setup, supported by the Area Under the Scaling Curve (AUSC) and Resolution Bias Score (RBS) metrics to measure model scalability allows less resource-intensive research, as it disentangles model size and general performance from scalability to image size and allows to solve this problem in isolation. We believe that this enables more researchers being able to contribute to this area in shorter time.

## B  ADDITIONAL EXPERIMENTAL RESULTS

### B.1  VLM'S PERFORMANCE DEGRADATION ON FULL BENCHMARK

In Figure 8 and Figure 8 we illustrate how performance of the VLMs degrade for three different positions of visual information in the white field.

Interestingly, Figure 8c illustrates effects of central crop being part of image pre-processing used in LLaVa-1.5. We recommend to always perform separate analysis of the position effects on the results before aggregating them.

### B.2  ASPECT RATIO ROBUSTNESS ANALYSIS

To complement our analysis of sensitivity to resolution and relative details size, we also analysed how VLMs react to the change of aspect ratio. We observed that for GPT-4o and Qwen-2.5, the aspect ratio does not present a significant challenge, while LLaVa-1.5's and LlaMa-3.2's performance is significantly affected.

Table 3: AUSC for different positions of visual cues with regard to aspect ratio

|           | Overall | Center | Top   | Bottom |
|-----------|---------|--------|-------|--------|
| LLaMa-3.2 | 0.752   | 0.755  | 0.738 | 0.763  |
| Qwen-2.5  | 0.839   | 0.839  | 0.835 | 0.843  |
| LLaVa-1.5 | 0.536   | 0.859  | 0.375 | 0.375  |
| GPT-4o    | 0.883   | 0.871  | 0.904 | 0.873  |

Interestingly, our analysis reveals that LLaVa1.5 is extremely sensitive to positioning of the visual cues when aspect ratio is imbalanced and strongly prefers critical information to be placed centrally.

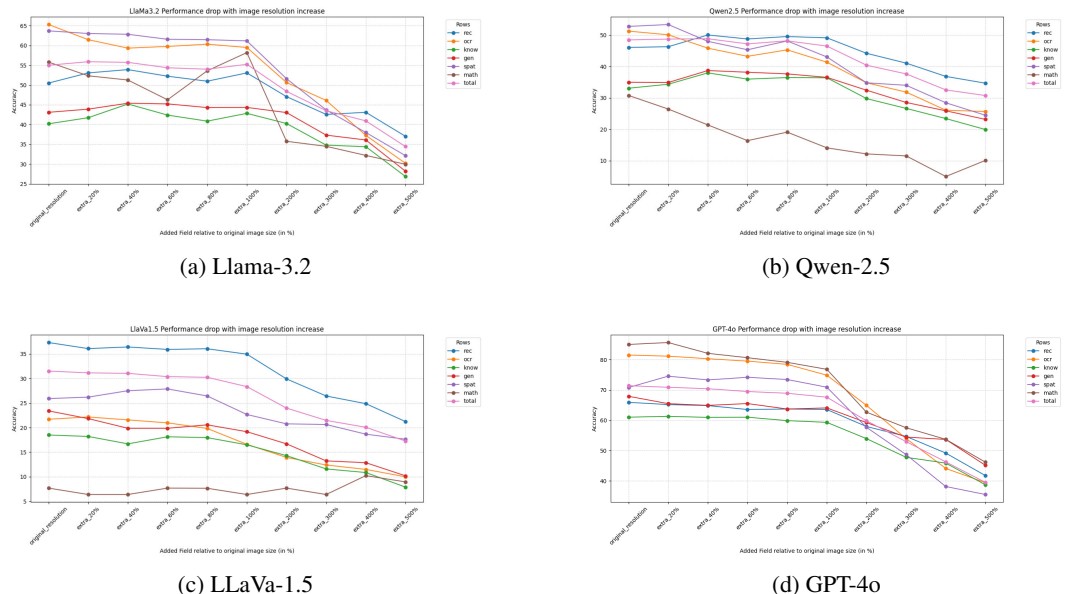

(a) Llama-3.2

(b) Qwen-2.5

(c) LLaVa-1.5

(d) GPT-4o

Figure 6: VLMs' sensitivity to image size increases across different categories of visual understanding (MMVet benchmark)

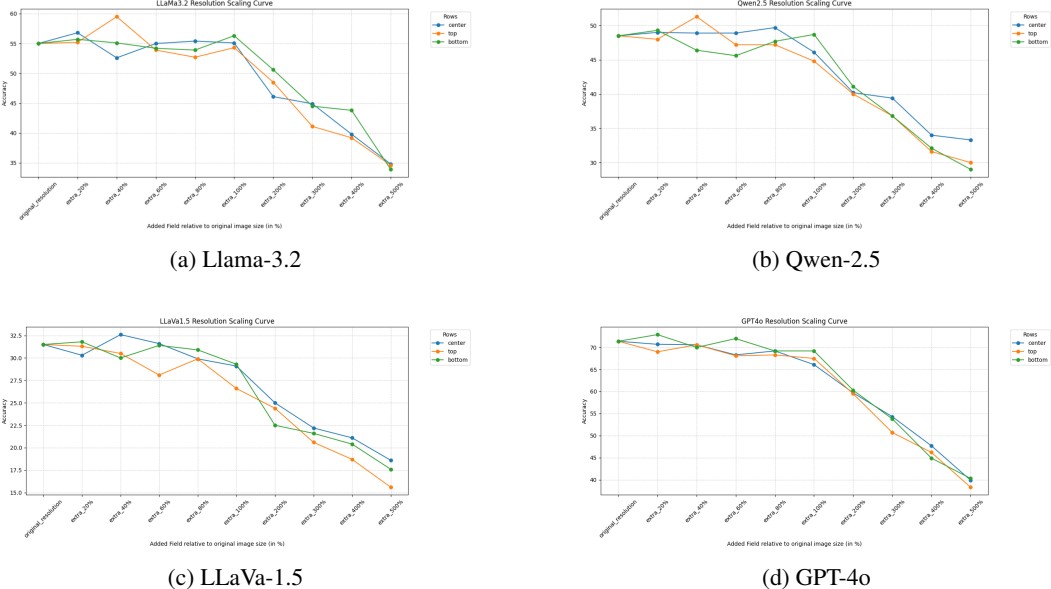

(a) Llama-3.2

(b) Qwen-2.5

(c) LLaVa-1.5

(d) GPT-4o

Figure 7: VLMs' sensitivity to image size increases for different positions of visual cues (MMVet benchmark)

### B.3 EXAMPLES OF CHANGE IN VLMS' ANSWERS WITH INCREASE OF ADDED WHITE MARGIN

In order to better understand the specifics of performance degradation with margin increase we manually reviewed several examples and present our results in this section (see Figures 10, 11, 12).

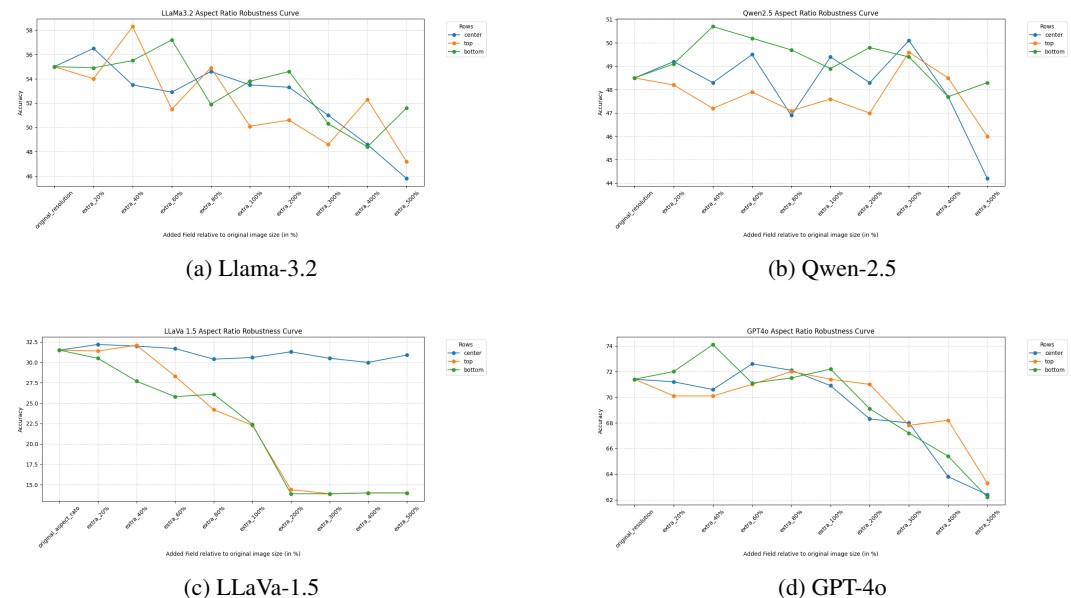

(a) Llama-3.2

(b) Qwen-2.5

(c) LLaVa-1.5

(d) GPT-4o

Figure 8: VLMs' robustness to the position of visual inputs for changed aspect ratios (MMVet benchmark)

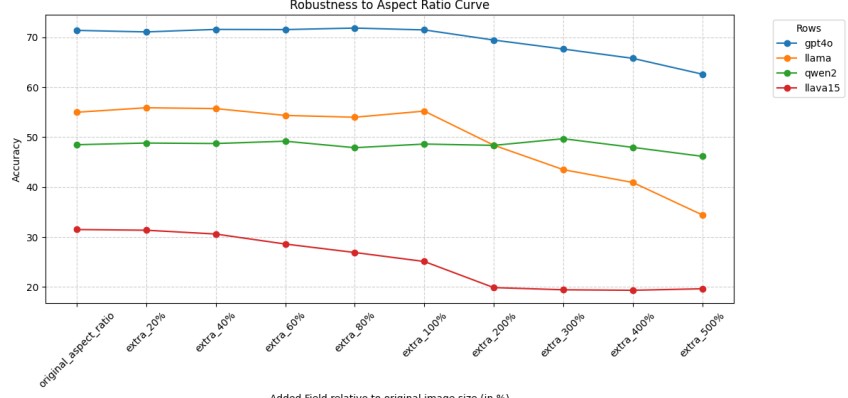

Figure 9: Plot of aspect ratio robustness as disbalance between image proportions increases (averaged across positions)

## C  COMPUTATIONAL BUDGET

All experiments involving open-source vision-language models (VLMs) were conducted using three NVIDIA RTX A6000 GPUs for inference. For benchmarking GPT-4o, we utilized the paid API access provided by OpenAI.

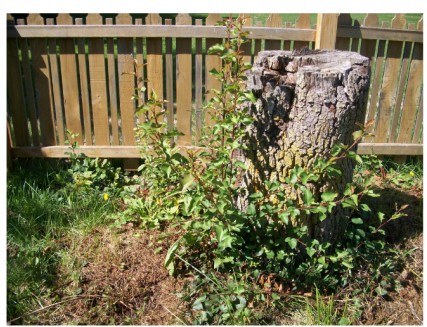

'What do both the stump and the fence have in common?'

Qwen-2.5

Answer with original resolution — "Both the stump and the fence have a wooden material as their primary component."

Answer with +100% margin — "Both the stump and the fence have a wooden material."

Answer with +200% margin — "Both the stump and the fence have a boundary or edge to them. The stump is the base of a tree that has been cut down, and the fence is a structure used to mark the boundary of a property. Both serve as markers for the edges of their respective areas."

Answer with +400% margin — "Both the stump and the fence have a vertical orientation."

Figure 10: Example of answer change with margin increase (Qwen-2.5)

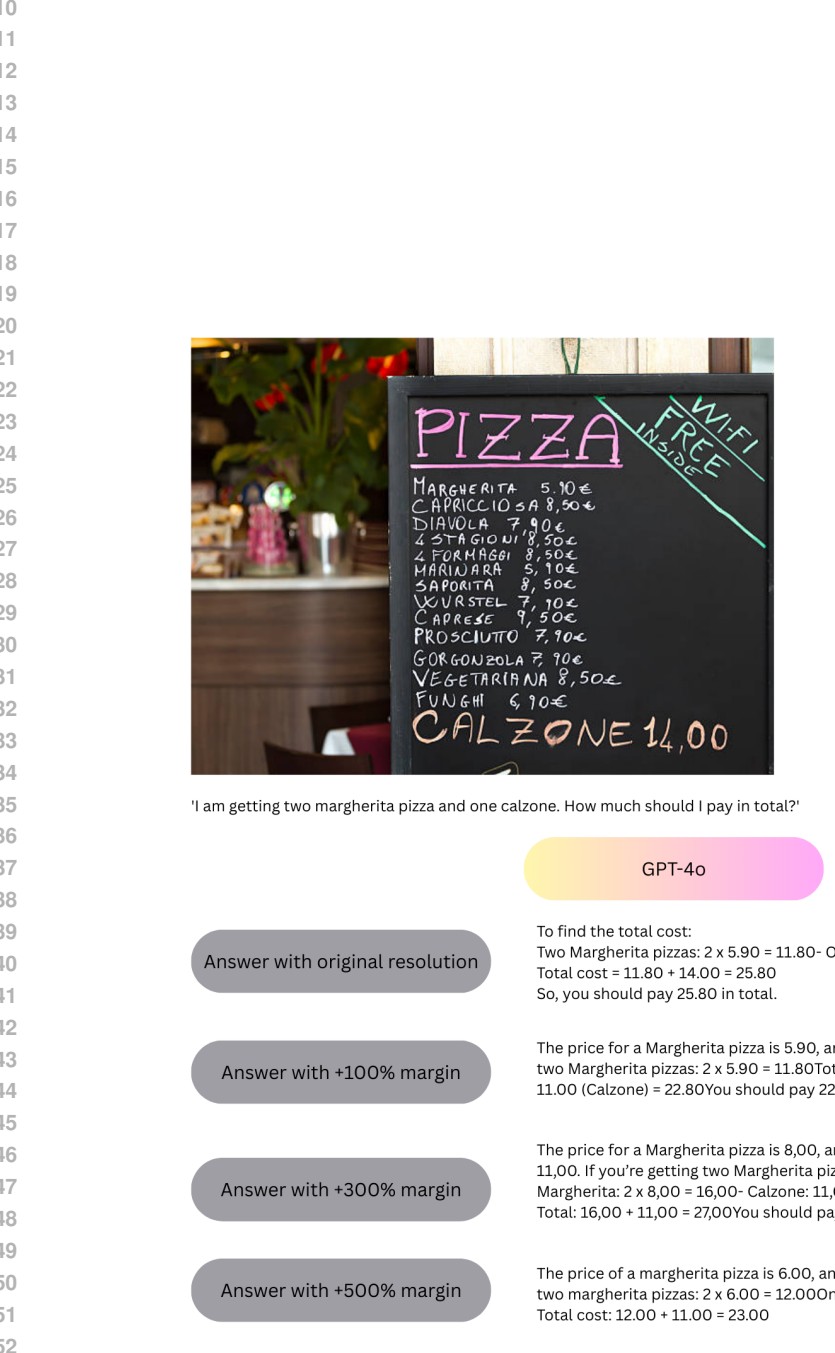

'I am getting two margherita pizza and one calzone. How much should I pay in total?'

GPT-4o

Answer with original resolution

To find the total cost:
Two Margherita pizzas: 2 x 5.90 = 11.80- One Calzone: 14.00
Total cost = 11.80 + 14.00 = 25.80
So, you should pay 25.80 in total.

Answer with +100% margin

The price for a Margherita pizza is 5.90, and a calzone is 11.00.For
two Margherita pizzas: 2 x 5.90 = 11.80Total: 11.80 (Margherita) +
11.00 (Calzone) = 22.80You should pay 22.80 in total

Answer with +300% margin

The price for a Margherita pizza is 8,00, and the Calzone is
11,00. If you're getting two Margherita pizzas and one Calzone:-
Margherita: 2 x 8,00 = 16,00- Calzone: 11,00
Total: 16,00 + 11,00 = 27,00You should pay 27,00 in total.

Answer with +500% margin

The price of a margherita pizza is 6.00, and a calzone is 11.00.For
two margherita pizzas: 2 x 6.00 = 12.00One calzone: 11.00
Total cost: 12.00 + 11.00 = 23.00

Figure 11: Example of answer change with margin increase (LLaMa-3.2)

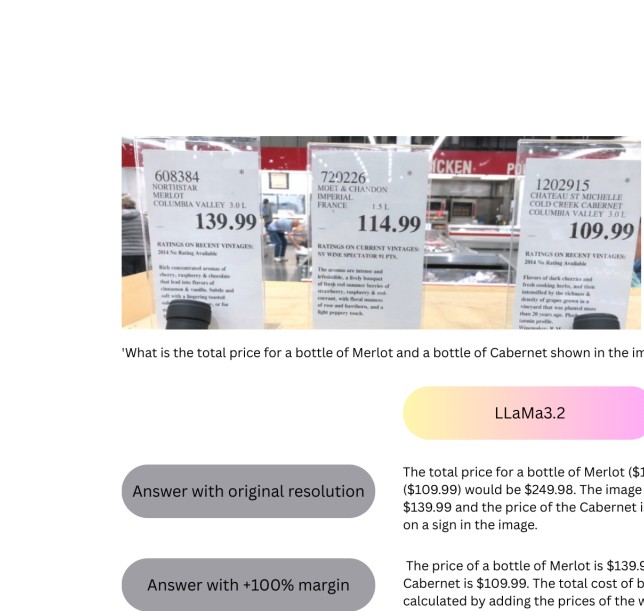

'What is the total price for a bottle of Merlot and a bottle of Cabernet shown in the image?'

LLaMa3.2

**Answer with original resolution**
The total price for a bottle of Merlot ($139.99) and a bottle of Cabernet ($109.99) would be $249.98. The image shows that the price of the Merlot is $139.99 and the price of the Cabernet is $109.99. The prices are displayed on a sign in the image.

**Answer with +100% margin**
The price of a bottle of Merlot is $139.99, and the price of a bottle of Cabernet is $109.99. The total cost of both wines is $249.98. This was calculated by adding the prices of the wines together: $139.99 + $109.99 = $249.98.

**Answer with +300% margin**
The total price for a bottle of Merlot and a bottle of Cabernet shown in the image is $213.98. The price of the Merlot is $139.99 and the price of the Cabernet is $73.99, so the total is $139.99 + $73.99 = $213.98.

**Answer with +500% margin**
The total price for a bottle of Merlot and a bottle of Cabernet shown in the image is $206.99. The Merlot is $139.99 and the Cabernet is $69.99. 139.99 + 69.99 = 206.98. 206.98 rounded up to the nearest whole number is 207.

GPT-4o

**Answer with original resolution**
The total price for a bottle of Northstar Merlot ($139.99) and a bottle of Chateau St Michelle Cold Creek Cabernet ($109.99) is $249.98.

**Answer with +100% margin**
The price for the Merlot from Columbia Valley is $139.99, and the price for the Cabernet from Columbia Valley is $109.99. The total price for both bottles is $249.98.

**Answer with +300% margin**
The total price for a bottle of Merlot ($139.99) and a bottle of Cabernet ($114.99) is $254.98.

**Answer with +500% margin**
"I'm **unable to view or recognize details in images, such as text or prices.** However, if you provide the prices for the Merlot and Cabernet, I can calculate the total for you."

Figure 12: Example of answer change with margin increase (LLaMa-3.2, GPT4o)

