# OpenReview forum: "EvalRes: Evaluating VLMs' Sensitivity to Image Resolution and Relative Detail Size"
_ICLR.cc/2026/Conference — Submitted to ICLR 2026_

### Official Review · Reviewer_nURN · 2025-10-31

**Soundness:** 3
**Presentation:** 2
**Contribution:** 1
**Rating:** 2
**Confidence:** 4

**Summary:**

The paper investigates the scalability and robustness of VLMs with respect to image resolution and the relative size of important visual details. The authors propose a controlled experimental setting that augments images with neutral margins to systematically test resolution and aspect ratio sensitivity, introducing two metrics — AUSC and RBS— as quantitative measures. Several popular VLMs are evaluated under different margin sizes, positions, and aspect ratios.

**Strengths:**

1. Controlled Benchmark Extension — The paper provides a simple yet reproducible way to alter benchmark datasets to isolate resolution and aspect ratio effects, which can be applied across existing VQA benchmarks.

2. New Metrics — AUSC and RBS are well-defined, intuitive metrics that allow decoupling resolution scalability from model size or overall accuracy.

3. Practical Implications — The results highlight non-trivial weaknesses in current VLM vision encoders, emphasizing the importance of resolution invariance in applications like OCR, mathematical reasoning, and diagram understanding.

**Weaknesses:**

1. Presentation & Naming Issues — There are multiple inconsistencies in formatting and nomenclature (e.g., table headings, capitalisation of “LLaVA”; missing model size specifications; incorrect naming such as “LlaVa-1.5” instead of “LLaVA-v1.5-XB”). These hinder clarity and professionalism.

2. Motivation May Be Weak — The proposed margin-based “resolution sensitivity” setting may not be crucial for next-generation models (e.g., o3-like multimodal LLMs) that can actively crop or resize relevant image regions on their own.

3. Limited Impact of Experimental Variations — In many cases (e.g., Tables 1 & 2), model performance changes across settings are small, suggesting that the proposed setup might not challenge stronger models significantly. The main large deviation is for LLaVA-v1.5, which is likely due to its center-cropping preprocessing rather than a fundamental resolution limitation.

4. Contribution Scope — The newly proposed metrics are essentially variants of existing evaluation measures (e.g., AUSC is conceptually an AUC-like metric), and the core experiments reveal only moderate performance differences.

5. No Deep Dive into Underlying Causes — While the paper reports performance degradation, there is limited deeper analysis (e.g., attention maps, reasoning chain analysis) to fully explain why such degradations occur.

**Questions:**

See weaknesses.

---

> ### Author Response · Authors · 2025-11-22
> **Response to Reviewer nURN**
>
> We thank the reviewer for the constructive assessment and for recognizing the controlled benchmark design. We address concerns below and will incorporate all suggested improvements in the updated version of the paper.
>
> W1. We appreciate the reviewer’s suggestions regarding model naming, headings, and formatting, and will ensure full consistency, add explicit model sizes in all tables, and revise figure/table formatting for clarity. These are straightforward changes that do not affect the results.
>
> W2. While advances in model architectures offer promising avenues for addressing resolution-related bottlenecks, it remains crucial to systematically test resolution and aspect ratio robustness and model scaling limits. We agree with the reviewers’ point that testing additional forms of image enhancement would be beneficial to better assess newer generations of models. Therefore, we add a set of experiments evaluating super-resolution while applying the AUSC and RBS metrics proposed earlier. Please find our preliminary results below:
>
> | Model                        | Original | Scale2 | Scale3 | Scale4 | AUSC  | RBS   |
> |-----------------------------:|--------:|------:|------:|------:|------:|------:|
> | Qwen2.5-VL-3B                | 0.656   | 0.555 | 0.399 | 0.298 | 0.689 | 0.173 |
> | Pixtral-12B                  | 0.610   | 0.569 | 0.573 | 0.541 | 0.898 | 0.118 |
> | Llava1.5-7B                  | 0.317   | 0.252 | 0.239 | 0.266 | 0.780 | 0.154 |
> | Llama3.2-Vision-Instruct-11B | 0.550   | 0.587 | 0.592 | 0.528 | 0.872 | 0.357 |
>
> Following this point, we also have extended the introduction section to provide additional motivation.
>
> W3. As demonstrated in Figures 3 and 4, even stronger open-source VLMs such as Qwen2.5 and Pixtral that are able to process images in native resolution as well as the commercial GPT-4o model suffer from significant accuracy degradation (e.g. Qwen2.5-VL-3B’s accuracy on the MMVet benchmark in original resolution is 48.9%, adding 200% white field decreases the performance by 9 absolute p.p. (accuracy 40.1%), and adding 500% white field causes a decrease by almost 20 absolute p.p. (accuracy 30.2%). An even more severe degradation is observed for the originally stronger GPT4o: 32 absolute p.p. (original accuracy 72%; accuracy with 500% white field 39%).
>
> W4. We agree that AUSC is by design similar to AUC-like measures. However, together with our controlled experimental design it serves a completely different purpose and allows for quantifying the ability of a model to maintain its performance when scaling to larger image sizes where relevant details become smaller relative to the image size (effectively longer visual context if no resizing is applied).
>
> W5. We agree with the reviewer and think that our research will benefit from the analysis of attention maps to better explain the reasons for the performance degradation. We are working on additional study of causes for performance degradation for a representative model (we chose Qwen2.5-VL-3B, as it is an open-source commonly used model that processes images in nearly native resolution) and will include it into the updated version of the paper.

---

### Official Review · Reviewer_6Hpo · 2025-10-31

**Soundness:** 2
**Presentation:** 3
**Contribution:** 2
**Rating:** 2
**Confidence:** 4

**Summary:**

The paper introduces EvalRes, a simple, controlled framework to evaluate VLMs’ sensitivity to image resolution and aspect ratio by padding each benchmark image with neutral margins while keeping the original content unchanged. It proposes two metrics: AUSC (Area Under the Scaling Curve) and RBS (Resolution Bias Score) to quantify scaling robustness and prediction instability under such transformations.

**Strengths:**

1. Clear, controlled manipulation: padding preserves answer-relevant pixels while varying overall image size, which would be proper for isolating preprocessing effects from reasoning.
2. Two concise metrics (AUSC/RBS) that separate robustness over items the model got right and the instability over items it got wrong.

**Weaknesses:**

1. Paper feels rushed: very short introduction with no references, formatting inconsistencies, and some mathematical formulations that do not clearly support new insights in the main text.
2. A fundamental concern I have is that the provided method does not actually change the signal resolution of the original scene. It simply adds pad/noise patches that alter packing/cropping and more specifically, input tokens of the vision model. This primarily would test padding/cropping sensitivity of the models rather than true resolution scalability.

**Questions:**

1. Following weakness #1, the paper seems a bit rushed e.g. (1) The introduction is very short and has no citations/references for the motivations (2) In figure 5, the plots and their titles do not have the same size and are not aligned (3) The metrics are relatively simple and therefore, the mathematical formulations do not seem to be necessary to be presented in the main text. Overall I think a few rounds of polishing might be necessary for the paper to become submission-ready.
2. To further elaborate on weakness #2: With the proposed approach, in my humble opinion, the paper is mostly testing how much the model can detect and ignore full and partial noise/pad tokens from its inputs, and does not analyze how much the model can handle different resolutions which contain different amount of high-frequency signal. In ViT-style encoders, white margins create (1) fully padded patches and (2) mixed patches (part margin, part content). This is not the same as feeding a genuinely higher-resolution image with more high-frequency signal and fine-grained information. In a proper study on this problem, one would try out different resolutions of the same images (so actually having more fine-grained signals as you increase the resolution) to test this. I would be interested to see such experiments to better validate the claims of the paper.
3. It is mentioned in section 3.1 that super-resolution (SR) models are (1) computationally expensive and (2) might add noise/irrelevant information to the image. Modern and older SR approaches such as GAN-based models like Real-ESRGAN [1] perform quite well and are publicly available on very small to large sizes. I would appreciate it if the authors can either update the claim or provide proper explanations, citations, or experiments to support it.
4. Several relevant works have not mentioned in the paper [2, 3, 4] which to some degree address the paper's questions. I would appreciate it if you could elaborate on these works and explain how they relate to the paper.
5. Could you elaborate on baseline selection? It seems there are additional baselines ([3, 4, 5]) that would be interesting to include or at least position against.

[1] Wang, Xintao, et al. "Real-esrgan: Training real-world blind super-resolution with pure synthetic data." _Proceedings of the IEEE/CVF international conference on computer vision_. 2021.

[2] Chai, Lucy, et al. "Any-resolution training for high-resolution image synthesis." _European conference on computer vision_. Cham: Springer Nature Switzerland, 2022.

[3] Xue, Le, et al. "xgen-mm (blip-3): A family of open large multimodal models." _arXiv preprint arXiv:2408.08872_ (2024).

[4] Wang, Peng, et al. "Qwen2-vl: Enhancing vision-language model's perception of the world at any resolution." _arXiv preprint arXiv:2409.12191_ (2024).

[5] Guo, Zonghao, et al. "Llava-uhd: an lmm perceiving any aspect ratio and high-resolution images." _European Conference on Computer Vision_. Cham: Springer Nature Switzerland, 2024.

---

> ### Author Response · Authors · 2025-11-22
> **Response to Reviewer 6Hpo**
>
> We thank the reviewer for the constructive critique and recognizing the strength of our work. We address concerns below.
>
> Q1. We thank the reviewer for the suggestions and will incorporate all mentioned points into the paper text. Specifically, we will (1) extend the introduction and provide additional motivation supported by relevant references, (2) consider shortening mathematical formulations for the metrics while moving the rest to the appendix, (3) improve the formatting in graphs and text.
>
> Q2/Q3. We agree that our method tests a specific aspect of resolution handling, but we argue that this is both intentional and valuable. What our method evaluates is resolution invariance—whether models maintain performance when the same semantic content occupies different pixel budgets within the fixed input resolution. This targeted evaluation eliminates confounding factors and  directly assesses:
> 1. Robustness to preprocessing artifacts (cropping, resizing)
> 2. Ability to locate and focus on relevant content regardless of its relative size
> 3. Sensitivity to content positioning and aspect ratio variations
>
> At the same time, we agree with the reviewer’s point, that it would be interesting to also see how super-resolution and downsizing of the image would affect the performance,  We have followed the reviewer’s suggestion of conducting additional experiments using the proposed Real-ESRGAN model. Please find our preliminary results below:
>
> | Model                        | Original | Scale2 | Scale3 | Scale4 | AUSC  | RBS   |
> |-----------------------------:|--------:|------:|------:|------:|------:|------:|
> | Qwen2.5-VL-3B                | 0.656   | 0.555 | 0.399 | 0.298 | 0.689 | 0.173 |
> | Pixtral-12B                  | 0.610   | 0.569 | 0.573 | 0.541 | 0.898 | 0.118 |
> | Llava1.5-7B                  | 0.317   | 0.252 | 0.239 | 0.266 | 0.780 | 0.154 |
> | Llama3.2-Vision-Instruct-11B | 0.550   | 0.587 | 0.592 | 0.528 | 0.872 | 0.357 |
>
> We will also update the corresponding claim as suggested by the reviewer in Q3.
>
> Q4. We will gladly incorporate additional citations to the mentioned references:
>
> [2] This paper focuses on image generation (synthesis), while VLM resolution-sensitivity concerns image understanding (perception). While there can be parallels, this research does not directly relate to the scope of our work.
>
> [3] We completely agree with the reviewer that BLIP3 is both relevant and different in architecture from the other models we have considered. We will include both the reference and additional experiments in the main paper.
>
> [4] Qwen2-VL and Qwen2.5-VL share many design choices. For illustrative purposes, we have chosen the newer Qwen2.5-VL model. However, we agree that the comparison would also be interesting and will include both reference and additional experiments in the main paper text.
>
> Q5. For the choice of baselines, we aimed to select the most commonly used modern VLMs that have differences in their image preprocessing steps (e.g., LlaVa1.5 still uses cropping, Qwen2.5-VL and Pixtral process images at native resolution, etc.) and design choices. This list can of course be extended to further models. We will include additional results in the camera ready version.

---

> > ### Comment · Reviewer_6Hpo · 2025-11-27
> >
> > Thanks for the rebuttal and for running the additional Real-ESRGAN experiments. I appreciate the clarifications and planned edits.
> >
> > In the new results, what do Scale2/3/4 columns exactly mean? Also, Pixtral and Llama-3.2-Vision-Instruct improve at some scales while the other two baselines do not. Could you please interpret this and elaborate on why this happens?
> >
> > Additionally, as discussed, although the recent experiments focus on the actual resolution sensitivity within a super-resolution pipeline, we all agree that the current protocol primarily analyzes position/relative-scale robustness and preprocessing sensitivity under fixed input size (i.e. fixed amount of information) and not general resolution sensitivity (i.e. different amount of signal within a scene), and I’m not fully convinced about the broader usefulness/generalizability of this narrower notion. Furthermore, I also agree with reviewer mKBi that for an empirical evaluation paper that mostly requires inference, the current experiments are not extensive enough, and additional baselines such as the ones I mentioned earlier could be interesting to try.
> >
> > Overall, given the rebuttal and added effort, I’m updating my score to weak reject. With additional experimentation with more baselines and a clearer motivation for the work, I would be leaning towards accepting the paper.

---

### Official Review · Reviewer_VNeL · 2025-10-31

**Soundness:** 2
**Presentation:** 2
**Contribution:** 1
**Rating:** 2
**Confidence:** 4

**Summary:**

This paper proposes a method to augment existing VLM benchmarks to varying image background sizes. The authors also proposes corresponding metrics and evaluates five VLMs on five benchmarks under their proposed setting.

**Strengths:**

* This paper demonstrates that some VLMs are sensitive to the position of images when placed on a white background.

**Weaknesses:**

* The technical contribution is overly incremental and trivial, offering little novel insight into VLM evaluation. Instead of providing an ad hoc data augmentation method, the authors ought to focus on a more fundamental question: given the multitude of possible ways to augment existing benchmarks, why is this specific metric or augmentation valuable over others? The authors must dive deeper into the essential reasons for their approach.
* Moreover, the paper does not achieve what it claims regarding increased evaluation difficulty or diversity. The proposed method simply places the original image onto white canvases of varying sizes. While this does generate more data, the underlying informative visual signals remain exactly the same and can be easily addressed by simple preprocessing steps (e.g., cropping the empty background). Consequently, the claimed benefit is not sound or robust.

**Questions:**

See above.

---

> ### Author Response · Authors · 2025-11-22
> **Response to Reviewer VNeL**
>
> We sincerely thank the reviewer for their feedback. While we respectfully disagree with several points, we appreciate the opportunity to clarify our contributions and address the core concerns about novelty and soundness.
>
> W1/W2.  With our experiments we aim to reveal a critical, yet previously underexplored failure mode: Our experiments show that state-of-the-art VLMs suffer from severe accuracy drops when the relevant content occupies different proportions of the input image due to differences in image resolution. This occurs despite the visual signal being identical and illustrated lack of invariance to this simple yet practically important transformation.
>
> The proposed framework does not aim to generate more data, but proposes a controllable, scalable and interpretable experiment set-up that allows quantifying model invariance to resolution of the original image, position of relevant visual cues, and their relevant size. While the augmentation method is simple, all tested models struggled with this set-up, not being able to detect relevant information.
>
> Following this feedback and suggestions from other reviewers, we will improve the quality and soundness of our work by introducing the following changes:
> 1. Extending introduction section to clarify the motivation and contributions of our method,
> 2. Additional experiments assessing model behaviour under super-resolution
> 3. Additional experiments with further models (including specialized resolution oriented models such as Llava-uhd and LlaVa-uhd v2, additional architectures such as BLIP3, etc.)
> 4. Additional in-depth analysis of the underlying reasons of performance drops, taking Qwen2.5-3B-VL as an example model. Providing analysis of the patch shifts in the image encoding stage and analysis of the attention maps.
>
> We thank the reviewer for the time invested into the review process and welcome any further questions!

---

### Official Review · Reviewer_mKBi · 2025-11-01

**Soundness:** 3
**Presentation:** 2
**Contribution:** 2
**Rating:** 2
**Confidence:** 4

**Summary:**

This paper presents a benchmark to evaluate VLMs' sensitivity to image resolution and relative finegrained detail, as well as two metrics for assesement. The authors propose a simple method to construct the benchmark by augmenting existing images with neutral backgrounds of varying sizes. Based on the benchmarking method, the authors propose two metrics, AUSC and RBS, to measure the VLMs' scalability with respect to resolution and aspect ratio, both metrics is interpretable. The authors evaluate several VLMs on the benchmark and provide analysis based on the evaluation results.

**Strengths:**

* The paper is easy to follow.
* The paper is well-motivated, as resolution is actually playing a very important role in vision-language models, yet few works have thoroughly investigated its impact.
* The proposed metrics are interpretable and make sense.
* The paper proposes two metrics, AUSC and RBS, to measure VLMs' scalability with respect to resolution and aspect ratio.

**Weaknesses:**

* In my view, the contribution of this paper is not significant enough, only proposes two metrics and an evaluation methodology, alough I acknowledge the task is well-motivated and important.
* The model versioning in the paper is confusing. Specifically, Qwen2.5 and Llama3.2 are text-only models, so using them to evaluate multimodal capabilities is weird, maybe what the authors mean is Qwen2.5VL?
* The experimental evaluation is insufficient; experiments were conducted on only five models, which is not convincing enough.

**Questions:**

* In L257, the recommended $|B_{correct}|$ should be greater than 75,  why, is there any explanation?
* Does adding margins around images alter the image domain, making them differ from natural images? Could this introduce a variable, as performance differences across models may stem not only from image resolution but also from whether their training data included similar images with margins?

---

> ### Author Response · Authors · 2025-11-22
> **Response to Reviewer mKBi**
>
> We sincerely thank the reviewer for their thoughtful feedback and for recognizing the motivation and interpretability of our work.
>
> Q1. The purpose of introducing a minimum size for ${B_{correct}}$ and ${B_{wrong}}$ is to ensure that downstream metrics—computed separately on correct and incorrect subsets—are based on samples large enough to yield stable and statistically interpretable estimates. A threshold of 75 observations is proposed as a pragmatic balance between statistical reliability and practical applicability across heterogeneous benchmarks.
>
> From a statistical perspective, CLT requires at least 30 samples, while the majority of empirical studies require about 100 samples for being able to make robust conclusions. 75 is a practical compromise - it is within the empirically used range and is not too restrictive, still allowing metric calculation on smaller specialized benchmarks taking weaker models into consideration.
>
> Given these goals, we chose 75 to make the evaluation of the considered models more comparable and the choice of benchmark more appropriate, as having both ${B_{correct}}$ and ${B_{wrong}}$​ at least 75 guarantees that the models are not compared on benchmarks that are either too easy (${B_{wrong}} \to 0$) or too challenging (${B_{correct}} \to 0$). The last case intuitively means that the model is producing answers close to random and scaling of its accuracy is not due to scaling capabilities, but purely based on chance.
>
> Q2. We thank the reviewer for this question and want to addressed mentioned concern from multiple perspectives:
> Margins and padding are not rare in VLM training data. We encounter numerous instances of natural margins, borders, and empty space in large-scale datasets (COCO, LAION) as well as established benchmarks (MMVet, CharXiv). Therefore, our augmentation approach does not introduce visual patterns that models have never encountered, but rather address naturally occurring phenomena
> Processing relevant content against a homogeneous background should conceptually be no more difficult than processing the same content surrounded by additional visual information that could act as distractors. Hence, our methodology explicitly eliminates confounding factors. Indeed, it tests a fundamental capability: identifying and reasoning about relevant content when it occupies varying proportions of the image canvas—a scenario that naturally occurs in real-world applications such as document analysis, UI screenshots, and images embedded in layouts.
>
> We argue that an inability to maintain performance on margin-augmented images reveals important model limitations, including: (1) preprocessing biases in different architectures (e.g., aggressive resizing that destroys fine-grained details), (2) insufficient robustness to spatial layout variations in training data, and (3) context length constraints when processing higher-resolution inputs.
> To further confirm that model performance degradation is not purely due to the training data distribution shift, but more fundamental design bottlenecks and context length constraints, we have run additional experiments with super-resolution.
>
> | Model                        | Original | Scale2 | Scale3 | Scale4 | AUSC  | RBS   |
> |-----------------------------:|--------:|------:|------:|------:|------:|------:|
> | Qwen2.5-VL-3B                | 0.656   | 0.555 | 0.399 | 0.298 | 0.689 | 0.173 |
> | Pixtral-12B                  | 0.610   | 0.569 | 0.573 | 0.541 | 0.898 | 0.118 |
> | Llava1.5-7B                  | 0.317   | 0.252 | 0.239 | 0.266 | 0.780 | 0.154 |
> | Llama3.2-Vision-Instruct-11B | 0.550   | 0.587 | 0.592 | 0.528 | 0.872 | 0.357 |
>
> Our preliminary results demonstrate that the performance of those models that do not perform resizing as a preprocessing step but process images at native resolution, drops with super-resolution as much as with added margin.
>
> W2. We appreciate the reviewer’s suggestions regarding model naming and will ensure full consistency, add explicit model sizes in all tables, and revise figure/table formatting for clarity. These are straightforward changes that do not affect the results.
>
> W3. We will add additional experiments with further models (including specialized resolution oriented models such as Llava-uhd and LlaVa-uhd v2, additional architectures such as BLIP3, etc.) in the updated version of this paper.

---

### Comment · Area_Chair_jh3n · 2025-11-25
**Discussion Period**

Dear Reviewers and Authors,

Thank you to the authors for submitting your rebuttal. We kindly encourage reviewers to take a moment to read the response and share any follow-up thoughts. Your timely engagement at this stage is highly valuable and helps ensure a fair, well-informed final decision.

We appreciate everyone’s efforts and contributions to the process.

Warm regards, Your AC

---

### Meta-Review · Area_Chair_NcQb · 2026-01-08

**Summary:**

The paper can be further improved from multiple aspects. For example, the paper did not analyze inconsistent improvement patterns across models; concerned about broader generalizability, and missing citations like Real-ESRGAN, BLIP3, Qwen2-VL, Llava-uhd, Any-resolution training papers. Whether the authors deliver on their promised additional experiments, especially more diverse model baselines and clearer motivation in the introduction is very important. Generally the paper is not ready to be accepted at the current form.

**Reviewer Concerns:**

The major concerns are the proposed method is ad hoc augmentation, and does not explain why this augmentation is valuable, also underlying visual signals remain unchanged, cropping empty background is not novel enough. Attention maps or reasoning chain analysis can be also beneficial. There are promised additional results expected, like more baselines, more model sizes, high-resolution models, etc.

**Reviewer Scores:**

Most reviewers did not engage in the discussion yet. Only 6Hpo's concerns were actively discussed and partially addressed. No one increased the scores.

---

### Decision · Program_Chairs · 2026-01-26

Reject